# Preventing Unintended Memorization by Covering with Over-Memorization

## Abstract

From the advances of deep learning, the privacy concerns of deep neural networks are in the limelight. A particular concern is privacy of the training data, which is often compromised by the model's inherent memorization capabilities. Suppressing such memorization can enhance privacy but introduces two main challenges: 1) removing a memorized instance from the training dataset will result in the model to memorize another instance instead, and 2) the memorization is essential for improving the generalization error. To address these challenges, we propose an over-memorization method that involves training the model with both the standard training set and a set of redundant, non-sensitive instances. Our method leverages the model's limited memorization capacity to focus on irrelevant data, thereby preventing it from memorizing the training data. Our empirical results demonstrate that this method not only enhances protection against membership inference attacks but also minimizes the loss of utility by effectively redirecting the model's generalization efforts towards non-sensitive instances.

## 1 Introduction

With the widespread success of deep neural networks across various fields, growing concerns have emerged regarding privacy violations, including the privacy of training data (Nasr et al., 2018; Shokri et al., 2017; Abadi et al., 2016; Dwork et al., 2014; Ye et al., 2024; Jagielski et al., 2024), intellectual property infringement by generative models (Kirchenbauer et al., 2023; Vyas et al., 2023; Smits & Borghuis, 2022), and ownership of the networks themselves (Maini et al., 2021; Liu et al., 2021; Kim et al., 2023). Among the various types of privacy for deep neural networks, the privacy of training data is particularly crucial, as it ensures that deep neural networks perform their tasks without compromising sensitive information.

Despite their capabilities, deep neural networks often inadvertently expose training data (Carlini et al., 2021; Nasr et al., 2023; Geiping et al., 2020), mainly due to the memorization of training instances (Carlini et al., 2022b; Zhang et al., 2023). Memorization, an inherent property of these networks, frequently occurs across a variety of deep learning tasks, from classification to generation (Feldman, 2020; Feldman & Zhang, 2020; Carlini et al., 2021; 2023). While memorization is essential for minimizing the generalization errors (Feldman, 2020; Feldman & Zhang, 2020), it also poses significant privacy risks. Notably, memorized instances can be exploited through membership inference attacks, which determine whether the data was used in training (Yeom et al., 2018; Shokri et al., 2017; Carlini et al., 2022a; Ye et al., 2022).

Addressing memorization directly such as regularization proves ineffective (Carlini et al., 2019), and simply removing memorized instances does not prevent networks from memorizing other data instances (Arpit et al., 2017). Therefore, reducing memorization without compromising other aspects of network performance is challenging. In response, we propose an innovative approach: rather than modifying the training algorithm, we aim to redirect the network's memorization capacity toward non-sensitive data, which poses no privacy risks. This method involves creating a dummy set—a collection of redundant, non-informative instances designed specifically to be memorized, thus preserving the utility of the network while protecting sensitive training data.

In this work, we propose an over-memorization method that trains the model with both the training set and a set of redundant, non-sensitive instances, which we named as *dummy set*. The dummy set is trained to effectively absorb the network's capacity to memorize, thereby improving the network in

terms of privacy. Also, the dummy set takes the role of reducing the generalization error, the network minimizes any loss of utility.

We evaluate our method on image classification and causal language modeling tasks, demonstrating enhanced privacy protection against membership inference attacks. Additionally, our analysis of memorization in networks trained with the over-memorization method confirms that this approach effectively mitigates privacy risks by significantly reducing the memorization of training data.

Our contributions are summarized as follows:

- We propose an over-memorization by training a set of non-sensitive instances to mitigate the challenges of reducing training data memorization.
- We validate the over-memorization method for the membership inference attacks, demonstrating significant improvements in privacy protection.
- We provide empirical evidence that our method minimizes the loss of utility by conducting a detailed analysis of its influences.

## 2 RELATED WORK

### 2.1 TRAINING DATA PRIVACY OF DEEP NEURAL NETWORKS

With the growing attention to deep learning and privacy, the privacy of training data has become a primary topic of discussion, as deep neural networks use vast amounts of data. Various works have explored building secure models to protect the training data privacy, including differential privacy (Dwork et al., 2014; Abadi et al., 2016; Li et al., 2021), auditing or understanding the individual influences (Jagielski et al., 2020; Ye et al., 2024), and federated learning (McMahan et al., 2017; Li et al., 2020; Truong et al., 2021). Although there are several strategies have been taken to protect training data violations, deep neural networks remain vulnerable to privacy threats such as membership inference (Fredrikson et al., 2014; Shokri et al., 2017; Carlini et al., 2022a; Ye et al., 2022), training data extraction (Carlini et al., 2021; Geiping et al., 2020), and jailbreaking attacks for large language models (Chao et al., 2023; Niu et al., 2024). In particular, membership inference has been widely studied for its application across a broad range of tasks, from the biomedical data and health records (Homer et al., 2008; Sankararaman et al., 2009; Backes et al., 2016; Zhang et al., 2022) to public data and large language models (Truex et al., 2019; Jagielski et al., 2024)m as well as various stereotypes (Yeom et al., 2018; Fredrikson et al., 2014; Sablayrolles et al., 2019; Choquette-Choo et al., 2021).

In this work, we focus on membership inference using reference models (Carlini et al., 2022a; Ye et al., 2022). Given a target model, an adversary trains reference models, each with a different training set drawn from the same data population. By analyzing the distribution of reference losses or confidence scores based on the membership status of each instance, an instance is considered a member of the target model's training data if it is likely to be drawn from the distribution of the training instance. This threat is closely related to the memorization of deep neural networks, by analyzing the memorization via influences (Feldman, 2020; Feldman & Zhang, 2020).

### 2.2 MEMORIZATION OF DEEP NEURAL NETWORKS

During training, deep neural networks memorize some of the training data, which is a distinct property from overfitting (Carlini et al., 2019; Feldman, 2020; Feldman & Zhang, 2020). Various approaches have demonstrated the aspects of memorization including the high capacity of deep neural networks (Zhang et al., 2017; 2021), over-parameterization (Daniely, 2020), and training dynamics (Stephenson et al., 2021). It has been demonstrated that rare and atypical samples in the training set are memorized, and these samples also contribute to generalization on unseen data (Feldman, 2020; Feldman & Zhang, 2020). Further, deep neural networks have sufficient capacity to memorize training samples even if the training samples are composed of random noise or labels (Zhang et al., 2017). However, deep neural networks often expose memorized data in response to privacy threats. Memorized instances are more susceptible to membership inference, as they are also easily memorized by other deep neural networks. Additionally, it is challenging to prevent memorization using well-known methods designed to alleviate overfitting (Carlini et al., 2019).

---

**Algorithm 1** Memorization score estimation in Feldman & Zhang (2020)

---

**Require:** training samples $\boldsymbol{Z} = \{\boldsymbol{z}_i\}_{i=1}^N$, learning algorithm $\mathcal{A}$, subset size $m$, number of trials $t$.

1: Sample $t$ random subsets of $\boldsymbol{Z} = \{\boldsymbol{z}_i\}_{i=1}^N$ of size $m$: $\boldsymbol{Z}_1, \boldsymbol{Z}_2, \ldots, \boldsymbol{Z}_t$.
2: Train model $h_k$ by running $\mathcal{A}$ on $\boldsymbol{Z}_k$ from $k = 1$ to $t$.
3: **for** $i = 1$ to $N$ **do**
4:  $\mathbf{Z}^+ = \{\boldsymbol{Z}_j : \boldsymbol{z}_i \in \boldsymbol{Z}_j, j = 1, \ldots, t\}$, $\mathbf{Z}^- = \{\boldsymbol{Z}_j : \boldsymbol{z}_i \notin \boldsymbol{Z}_j, j = 1, \ldots, t\}$
5:  $\widetilde{\mathrm{mem}}(\boldsymbol{Z}, \boldsymbol{z}_i) := \mathbb{E}_{\mathbf{Z}^+} \ell_h(z_i; \theta) - \mathbb{E}_{\mathbf{Z}^-} \ell_h(z_i; \theta)$
6: **end for**
7: **return** $\widetilde{\mathrm{mem}}(\boldsymbol{Z}, z_i)$ for all $i = 1, \ldots, N$.

---

Instead of preventing memorization by externally modifying the training algorithm, we embrace the memorization inherent in deep neural networks and manage it in a novel way. We construct a set of redundant samples, which we name the *dummy set*, and train deep neural networks using both the given training set and the dummy set. Due to the limited capacity for memorization, deep neural networks memorize less of the actual training data when trained with the addition of a dummy set compared to training without it. Furthermore, we carefully train with the dummy set, aiming to minimize the loss of utility.

## 3 METHOD

In this section, we present the concept of memorization used in our approach and introduce a method to prevent the memorization of training data by employing dummy sets.

### 3.1 QUANTIFYING MEMORIZATION VIA INFLUENCES

Given a training algorithm $\mathcal{A}(\cdot)$, a training set $\boldsymbol{S}$ drawn from a data population $\mathcal{P}$, a parametric model $h_\theta$ parameterized by $\theta$, and a loss measure $\ell_h(\boldsymbol{z}; \theta)$, the influence score of given instances from $\boldsymbol{z}$ to $\boldsymbol{z}'$ is formally defined as:[1]

$$\mathrm{infl}(\boldsymbol{S}, \boldsymbol{z}, \boldsymbol{z}') := \mathbb{E}_{\theta \leftarrow \mathcal{A}(\boldsymbol{S})} \ell_h(\boldsymbol{z}'; \theta) - \mathbb{E}_{\theta \leftarrow \mathcal{A}(\boldsymbol{S} \setminus \{\boldsymbol{z}\})} \ell_h(\boldsymbol{z}'; \theta), \tag{1}$$

where $\theta \leftarrow \mathcal{A}(\boldsymbol{S})$ indicates that the parameter $\theta$ is the result of $\mathcal{A}(\cdot)$ and $\boldsymbol{S}$. For classification tasks, memorization is defined through self-influence, denoted as $\mathrm{mem}(\boldsymbol{S}, \boldsymbol{z}) := \mathrm{infl}(\boldsymbol{S}, \boldsymbol{z}, \boldsymbol{z})$. Since the training algorithm $\mathcal{A}(\cdot)$ remains unchanged throughout this paper, we omit it from eq. 1 for the sake of simplicity.

We expand the concept of memorization and influence from individual instances $\boldsymbol{z}$ and $\boldsymbol{z}'$ to sets of instances, denoted as $\boldsymbol{Z}$ and $\boldsymbol{Z}'$. For two sets, $\boldsymbol{Z} = \{\boldsymbol{z}_i\}_{i=1}^{|\boldsymbol{Z}|}$ and $\boldsymbol{Z}' = \{\boldsymbol{z}_j'\}_{j=1}^{|\boldsymbol{Z}'|}$, we define the influence score over sets as:

$$\mathrm{infl}(\boldsymbol{S}, \boldsymbol{Z}, \boldsymbol{Z}') := \sum_{j=1}^{|\boldsymbol{Z}'|} \left( \mathbb{E}_{\theta \leftarrow \mathcal{A}(\boldsymbol{S})} \ell_h(\boldsymbol{z}_j'; \theta) - \mathbb{E}_{\theta \leftarrow \mathcal{A}(\boldsymbol{S} \setminus \boldsymbol{Z})} \ell_h(\boldsymbol{z}_j'; \theta) \right). \tag{2}$$

and as same as memorization score of single instance $\boldsymbol{z}$, the memorization score for a set $\boldsymbol{Z}$, analogous to the single-instance case, is defined as $\mathrm{mem}(\boldsymbol{S}, \boldsymbol{Z}) := \mathrm{infl}(\boldsymbol{S}, \boldsymbol{Z}, \boldsymbol{Z})$. Using the estimator outlined in Algorithm 1, proposed by Feldman & Zhang (2020), we can estimate both memorization and influence scores by training multiple models on different training sets.

Since the memorization score is defined as the difference between expected losses, it is inherently dependent on the training set $\boldsymbol{S}$. For instance, if the training set contains a sufficient number of informative instances that help reduce generalization error, the memorization score for each individual instance tends to be relatively low. To validate this, we train 100 CIFAR-10 models using different training sets, with the training set size varying from 5000 to 45000. We then estimated the memorization score following alg. 1 and plotted the scores alongside the corresponding test accuracies.

---

[1] In (Feldman & Zhang, 2020), influence score is suggested by comparing the accuracy over the models with given instance and without counterpart.

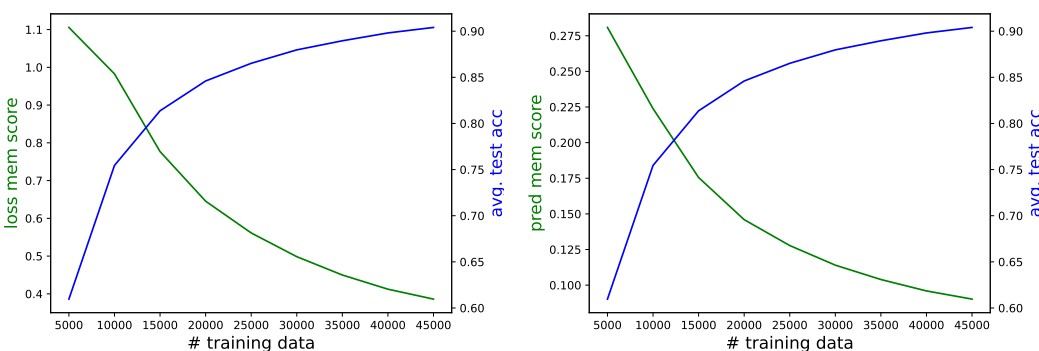

Figure 1: Memorization score while varying number of training data. **(Left)** memorization score with losses, **(Right)** memorization score with predictions (accuracies).

As shown in Fig. 1, increasing the number of training samples reduces both the generalization error and the average memorization score per training sample. Although some instances are inevitably memorized, increasing the training set size can reduce the overall memorization score by distributing the model's memorization across a broader set of data points.

### 3.2 USING ADDITIONAL TRAINING SET TO REDUCE MEMORIZATION

Since memorization is essential for reducing generalization error (Feldman & Zhang, 2020) and is difficult to avoid (Carlini et al., 2022c), we acknowledge these characteristics and propose leveraging the memorization capacity of deep neural networks by training on redundant, meaningless set of samples. This approach aims to occupy the network's memorization capacity while minimizing the memorization of sensitive data. To begin, we investigate the effect of training sets on memorization and influence using the formulation in eq. 1. Our setup involves a training set $S_t$ sampled from the target data population and a set of redundant samples $S_d$. Our goal is to reduce the memorization score $\texttt{mem}(S_t, z_t)$ for each training sample $z_t$ by training with the combined set of $S_t$ and the dummy set $S_d$, denoted as $\texttt{mem}(S_t \cup S_d, z_t)$. The difference between $\texttt{mem}(S_t \cup S_d, z_t)$ and $\texttt{mem}(S_t, z_t)$ can be expressed as[2]:

$$\texttt{mem}(S_t \cup S_d, z_t) - \texttt{mem}(S_t, z_t) = \texttt{infl}(S_t \cup S_d, S_d, z_t) - \texttt{infl}(S_t \cup S_d \backslash \{z_t\}, S_d, z_t). \quad (3)$$

This equation highlights that regardless of whether $S_d$ is sampled from the same data population as $S_t$, the change in the memorization score is influenced by two terms: $\texttt{infl}(S_t \cup S_d, S_d, z_t)$ and $\texttt{infl}(S_t \cup S_d \setminus z_t, S_d, z_t)$. For the first influence term, $\texttt{infl}(S_t \cup S_d, S_d, z_t)$, we expect it to remain relatively stable, as training with the combined set $S_t \cup S_d$ already includes $z_t$, and thus adding $S_d$ should not cause a significant change. Our primary objective is to maximize the second influence term, $\texttt{infl}(S_t \cup S_d \setminus z_t, S_d, z_t)$, which implies that training should generalize well for $z_t$ when we train with a superset composed of a different training set and the dummy set $S_d$.

As a result, we identify the key properties of the dummy set that are essential for effectively reducing memorization in the training set. First, the dummy set should enhance model performance after training. When training with the superset of the training set $S_t \setminus \{z_t\}$ and the dummy set $S_d$, the model reduces generalization error, thereby increasing the influence term $\texttt{infl}(S_t \cup S_d \setminus z_t, S_d, z_t)$. Second, each dummy sample $z_d$ within the dummy set $S_d$ should minimize correlation with individual training samples $z_t$. If $z_d$ contains information that is correlated with $z_t$, the model risks inadvertently memorizing and leaking information about $z_t$ through $z_d$.

### 3.3 COVERING UNINTENDED MEMORIZATION BY MEMORIZING DUMMY SET

Let us outline how to construct the dummy set and train the dummy set to reduce the memorization of training samples. We begin with the empirical risk minimization using stochastic gradient descent

---

[2]The detailed derivation is provided in the appendix.

---

**Algorithm 2** Training dummy set

---

**Require:** training algorithm $\mathcal{A}$, model $\tilde{h}_{\tilde{\theta}}$, initialized dummy set $\boldsymbol{S}_d$, training set $\boldsymbol{S}_t$, learning rate $\eta_h$ for model annd $\eta_d$ for dummy set.

1: Initialize the dummy set $\boldsymbol{S}_d$ and $\tilde{h}_{\tilde{\theta}}$ as random.
2: **while** converge **do**
3:     Sample training batch $B_t$ and dummy batch $B_d$ from $\boldsymbol{S}_t$ and $\boldsymbol{S}_d$.
4:     $\tilde{\theta} := \tilde{\theta} - \eta_h/(|B_t| + |B_d|)(\sum_{\boldsymbol{z}_t \in B_t} \nabla_{\tilde{\theta}} \ell(\boldsymbol{z}_t; \tilde{\theta}) + \sum_{\boldsymbol{z}_d \in B_d} \nabla_{\tilde{\theta}} \ell(\boldsymbol{z}_d; \tilde{\theta}))$
5:     $B_d := B_d - \eta_d/(|B_t| + |B_d|) \sum_{\boldsymbol{z}_d \in B_d} \nabla_{\boldsymbol{z}_d} \ell(\boldsymbol{z}_d; \tilde{\theta})$
6: **end while**
7: **return** $\boldsymbol{S}_d$

---

(SGD) which is widely used for training the model $h$. One iteration of stochastic gradient descent can be written as,

$$\theta_{\text{new}} := \theta_{\text{prev}} - \frac{\eta}{|B_t|} \sum_{\boldsymbol{z}_t \in B_t} \nabla_\theta \ell(\boldsymbol{z}_t; \theta_{\text{prev}}), B_t \sim \boldsymbol{S}_t \tag{4}$$

where $\ell$ denotes the loss measure for the given objective, and $\theta_{\text{new}}$ and $\theta_{\text{prev}}$ represent the previous and after parameters in SGD. The training batch $B$ is sampled from the training set $\boldsymbol{S}_t$ and $\eta$ is the learning rate. Our objective is to reduce memorization by training with the dummy set $\boldsymbol{S}_d$ without modifying the training algorithm (SGD). Since our method does not alter the training algorithm, one training step involving the dummy set can be expressed as follows:

$$\hat{\theta}_{\text{new}} := \hat{\theta}_{\text{prev}} - \frac{\eta}{|B_t| + |B_d|} \left[ \sum_{\boldsymbol{z}_t \in B_t} \nabla_{\hat{\theta}} \ell(\boldsymbol{z}_t; \hat{\theta}_{\text{prev}}) + \sum_{\boldsymbol{z}_d \in B_d} \nabla_{\hat{\theta}} \ell(\boldsymbol{z}_d; \hat{\theta}_{\text{prev}}) \right], B_t \sim \boldsymbol{S}_t, B_d \sim \boldsymbol{S}_d, \tag{5}$$

where $\boldsymbol{z}_d$ denotes a dummy sample from the dummy batch $B_d$.

Recall the dummy set should satisfy two conditions: 1) the dummy set should reduce generalization error, and 2) it contains minimal evidence of the training set $\boldsymbol{S}_t$. The second condition can be fulfilled by constructing the dummy set with randomized, noisy samples. For instance, in the context of image classification, dummy samples can be generated using randomized gaussian noise and randomized soft labels that have zero correlation with the classification objective. However, while this approach helps ensure minimal evidence, it does not inherently contribute to reducing generalization error in general. Thus, we *train* the dummy set to improve generalization performance. We begin by initializing the dummy samples as randomized images and soft labels. To optimize the dummy set, we create a separate model specifically for training the dummy set. Using the initialized model and the dummy set, we apply coordinate descent for both the model and the dummy set to minimize the task objective, as illustrated in Algorithm 2.

### 3.4 How to construct dummy set for each task

The introduced dummy set can take various forms, making it applicable to a wide range of tasks, from image classification to language modeling. In this section, we present the specific form of the dummy set utilized in our experiments.

**Image classification** In the image classification task, each data point consists of an image and its corresponding label. Each initialized dummy image has the same dimensionality of the training images and has pixel intensity values ranging from 0 to 1. The corresponding soft labels are also initialized randomly. During training, we do not restrict the values of the tensors to remain within the 0 to 1 range, which means the resulting tensors may exceed this range and are not visualized due to their potentially unbounded values.

**Language modeling** The objective of language modeling is to predict the next token based on a given sequence of previous tokens. We define the dummy tokens as soft tokens, each initialized as a convex combination of tokens typically used in language models. In each single dummy token, the candidate tokens for this convex combination are randomly selected before training the dummy set, and only these candidate tokens are utilized during the training process. As a result, the dummy sequence consists of a randomly selected convex combination of tokens. This approach, which

Table 1: Memorization score for CIFAR-10 and CIFAR-100 datasets. Training with dummy set from alg. 2 helps to reduce average memorization over training set.

| Dataset | training type | mem-**loss.** | mem-**pred.** |
|---|---|---|---|
| **CIFAR-10** | without dummy | $0.454 \pm 1.224$ | $0.105 \pm 0.224$ |
| | random dummy | $0.458 \pm 1.212$ | $0.105 \pm 0.221$ |
| | **trained dummy** | $\mathbf{0.441 \pm 1.054}$ | $\mathbf{0.104 \pm 0.221}$ |
| **CIFAR-100** | without dummy | $1.740 \pm 2.272$ | $0.376 \pm 0.368$ |
| | random dummy | $1.743 \pm 2.242$ | $0.379 \pm 0.363$ |
| | **trained dummy** | $\mathbf{1.684 \pm 2.181}$ | $\mathbf{0.355 \pm 0.359}$ |

employs sparse token sequences instead of dummy embeddings, effectively reduces both the memory and computational budget. We can control a ratio of candidate tokens that are used for convex combination and all tokens used for the language model, which we call a sparsity. The larger sparsity gives a high degree of freedom while training the dummy set but requires a larger computational cost. This is the difference with the dummy set of image classification, that the dummy has less potential to reduce the memorization while losing the generalization performance. However, the sparse dummy sequence still effectively reduces the memorization of the training set.

## 4 EXPERIMENTS

In this section, we present empirical results comparing the memorization effects of standard empirical risk minimization with our method of training using dummy sets to mitigate the memorization of training data.

### 4.1 SETUP

Our experiments are conducted on two tasks: image classification using CIFAR-10 and CIFAR-100, and causal language modeling with Wikitext-103 (Merity et al., 2016).

**CIFAR-10,100** The CIFAR-10 and CIFAR-100 datasets each contain 50,000 training samples. For measuring memorization scores and membership inference, we prepare subsets of size 25,000. For all image classification models, we employ the ResNet-18 architecture. Each model is trained using the minibatch SGD optimizer for 100 epochs, with a consistent batch sampling rate of 0.004, resulting in a training batch size of 100. When incorporating the dummy set into the training process, the sampling rate for the dummy set remains fixed at 0.004 to maintain consistency in the training algorithm. For example, if the dummy set size is set to 5,000, the dummy batch size will be 20. We utilize Adam optimizer to train the dummy set in Algorithm 2.

**Wikitext-103** It is common to utilize pretrained language models and fine-tune them using a specific training set. In our causal language modeling experiment, we aim to demonstrate that our method, which employs a dummy set, is effective in mitigating memorization over the fine-tuning dataset. To this end, while using the Wikitext-103 dataset, we divide it into 1,000 chunks, with each model trained on 100 chunks, resulting in approximately 180,135 sentences per model. We employ Pythia (Biderman et al., 2023)-70m model for our causal language modeling experiments. Each model is fine-tuned for 3 epochs without gradient accumulation. For the dummy set in causal language modeling, we ensure that each dummy set contains one-fifth of the token length of the training set. Prior to training the dummy set, we initialize it as a sparse soft token sequence, as described in sec. 3.4, and focus solely on training these sparse token sequences. During the training phase with the dummy set, we concatenate the randomly sampled dummy sequences to the sampled training token sequences and perform gradient descent on the concatenated sequences.

### 4.2 MEMORIZATION SCORE COMPARISON

We estimate the memorization score following alg. 1 to assess the effectiveness of our proposal–training with dummy sets. We estimate the memorization score for CIFAR-10 and 100 datasets following alg. 1 (Feldman & Zhang, 2020). We train a total of 400 models using a training set size of

Table 2: Results on CIFAR-10 dataset. We provide both accuracy on CIFAR-10 and loss over the dummy set. All the models with over-memorization show lower AUROC against LiRA attack.

| Metric | Standard | Over-memorization (# of dummies) | | |
|---|---|---|---|---|
| | | 1000 | 5000 | 25000 |
| Acc (%) | $87.58 \pm 0.11$ | $87.13 \pm 0.09$ | $\mathbf{87.61 \pm 0.23}$ | $86.07 \pm 0.13$ |
| Dummy loss | - | $.0002 \pm .0001$ | $.0002 \pm .0001$ | $.0008 \pm .0003$ |
| **Results on Membership inference** | | | | |
| AUROC | $.6373 \pm .0041$ | $.6241 \pm .0013$ | $.6100 \pm .0027$ | $.5995 \pm .0045$ |
| AUROC Diff. | - | $.0132 \pm .0054$ | $.0273 \pm .0067$ | $\mathbf{.0378 \pm .0086}$ |

Table 3: Results on Wikitext-103 datasets. We provide the perplexity (PPL) for both the Wikitext-103 dataset and the dummy set. All the over-memorized models protect more Wikitext-103 samples than standard training.

| Metric | Standard | Over-memorization (sparsity of soft tokens) | | |
|---|---|---|---|---|
| | | sparsity=1e-4 | sparsity=3e-4 | sparsity=5e-4 |
| PPL | $\mathbf{20.13 \pm 0.04}$ | $20.60 \pm 0.13$ | $20.41 \pm 0.09$ | $20.15 \pm 0.06$ |
| Dummy PPL | - | $26.87 \pm 7.61$ | $816.69 \pm 13.31$ | $3033.82 \pm 30.14$ |
| **Results on Membership inference** | | | | |
| AUROC | $.9688 \pm .0021$ | $\mathbf{.7972 \pm .0041}$ | $.8007 \pm .0029$ | $.7996 \pm .0132$ |
| AUROC Diff. | - | $\mathbf{.1716 \pm .0033}$ | $.1681 \pm .0044$ | $.1692 \pm .00103$ |

25,000. Our comparisons involve three scenarios: standard empirical risk minimization, training with randomly initialized dummy sets, and training with trained dummy sets as outlined in alg. 2. Each dummy set utilized during training contains 5,000 samples. The results are presented in Table 1.

We measure the memorization score based on both loss (mem-**loss**) and prediction (mem-**pred**). The results indicate that, in all cases, the trained dummy set contributes to a reduction in the average memorization score over the training set. Conversely, when models are trained with random noise as the dummy set, they often exhibit a higher generalization error compared to models trained without any dummy set. This occurs because the random noise dummy set, which is not trained using alg. 2, fails to effectively reduce the generalization error and thus does not contribute to proper memorization. These findings suggest that training with the dummy set and utilizing trained dummies significantly aids in lowering the average memorization scores across the training set.

## 4.3 RESULTS ON MEMBERSHIP INFERENCE

We conduct membership inference for both image classification and causal language modeling tasks, comparing models trained with and without dummy sets. Membership inference aims to determine whether a given instance is part of the training set of the target model. Among various membership inference methods, we adopt the approach that utilizes reference models as described in Carlini et al. (2022a); Ye et al. (2022). This process involves several steps: first, we train reference models, each using a training set sampled from the same data population as the target model. Next, we compute the losses for the given target samples. By comparing the losses from the target model with those from the reference models, we can establish a threshold to predict whether the target data was included in the training set. Using different thresholds allows us to plot the receiver operating characteristic (ROC) curve and compare the area under the curve (AUROC) to assess the model's vulnerability to membership inference. This method of membership inference with reference models is closely tied to the memorization properties of deep neural networks, as it involves comparing losses across various combinations of training sets, thereby making the AUROC values highly correlated with the degree of memorization.

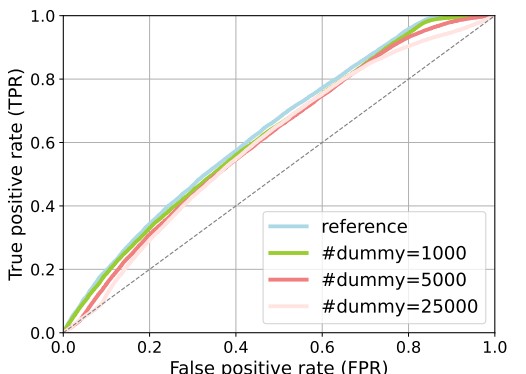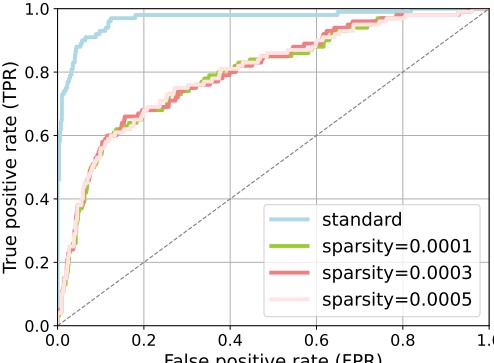

Figure 2: Resulting ROC curves for membership inference on CIFAR-10 **(left)** and Wikitext-103 **(right)**. The result shows that training with the dummy set helps confuse the membership status of the training instances.

Table 4: Membership inference results on dummy variants. **(Left)** Results on using subset of the pretrained dummy sets size of 5000 for CIFAR-10. **(Right)** Results on dummy sets where each dummy image in the dummy set is partially trained.

| Dummy partition | AUROC | Acc. |
|---|---|---|
| Full (#dummy=5000) | $.6100 \pm .0027$ | $87.61 \pm 0.23$ |
| half (1/2) | $.6185 \pm 0.0014$ | $87.58 \pm 0.15$ |
| one third (1/3) | $.6191 \pm 0.0004$ | $87.19 \pm 0.16$ |
| quarter (1/4) | $.6212 \pm 0.0029$ | $87.59 \pm 0.40$ |

| Sparsity for dummy | AUROC | Acc. |
|---|---|---|
| Dense (#dummy=5000) | $.6100 \pm .0027$ | $87.61 \pm 0.23$ |
| sparsity=0.3 | $.6131 \pm .0023$ | $87.28 \pm 0.15$ |
| sparsity=0.5 | $.6089 \pm .0032$ | $88.02 \pm 0.20$ |
| sparsity=0.7 | $.6156 \pm .0015$ | $87.56 \pm 0.13$ |

To conduct membership inference, we train 400 reference models for the CIFAR-10 dataset and 100 reference models for the Wikitext-103 dataset. The reference models utilize a training set size of 25,000 samples for image classification and 10% of the token length of the Wikitext-103 training set for causal language modeling. To assess robustness against membership inference, we perform experiments across various hyperparameter settings, including dummy set sizes for image classification and sparsity levels that define the number of learnable tokens for causal language modeling. We conduct three experiments for each case, and the results are presented in Tables 2 and 3.

In the image classification experiments, we varied the size of the dummy set, testing sizes of 1,000, 5,000, and 25,000. The dummy set effectively reduces the memorization of the training set without compromising the generalization error. Notably, the size of the dummy set does not significantly impact either the generalization error or the memorization of the training set. Since the training algorithm maintains a consistent batch sampling rate across all dummy set sizes, the influence of individual dummy samples is diminished. In contrast, the results from the causal language modeling experiments indicate that the sparsity of the dummy set is critical for mitigating memorization. A lower sparsity level restricts the degree of freedom within the dummy set, resulting in increased memorization. Overall, in all cases utilizing the dummy set, we observed a lower area under the curve (AUROC) in membership inference tasks, indicating that the model exhibits reduced memorization of the training set.

### 4.4 ANALYSIS ON DUMMY SET VARIANTS

We created variants of dummy sets, including subsets derived from pretrained dummy sets and sparsified image dummies.

**Using a subset of dummy sets**    From the trained dummy set, we divided it into subsets of sizes 2, 3, and 4, and subsequently trained the image classification model using these split portions. The results are presented on the left side of Table 4. Since the training of the dummy set is specifically tailored to reduce memorization in the target training set, the subsets do not achieve the same level

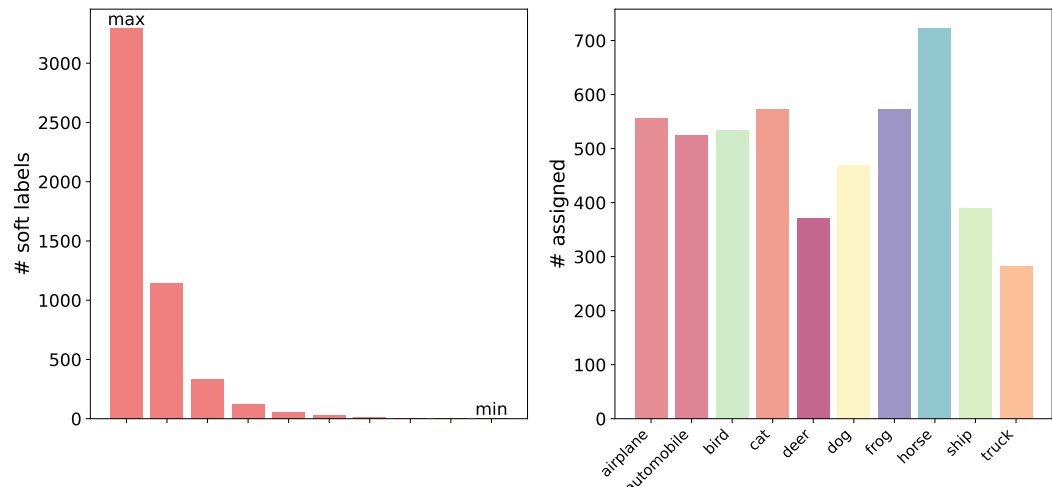

Figure 3: Assigned labels for dummy set. **(Left)** number of occurrences of each label index. **(Right)** average number of occurrences in CIFAR-10 dummy set.

of memorization reduction as the full dummy set. Notably, the quarter-sized subset, which contains 1,250 dummy samples, exhibits lower AUROC performance than the dummy set with just 1,000 samples, as shown in Table 2. This indicates that the total number of dummy samples is crucial for effectively reducing memorization over the training sets. However, it is important to note that larger dummy sets may also lead to increased generalization error, making the careful selection of dummy set size essential for minimizing memorization.

**Dummy set with sparse image**   We conducted experiments using sparse dummy sets in causal language modeling to address training costs. To evaluate the impact of sparse dummies, we also performed experiments on image classification tasks incorporating sparse dummy sets. The results on the right side of Table 4 indicate that sparse dummy sets consistently underperform compared to their dense counterparts. The sparse dummies exert a lesser influence on the classification model than the dense ones, resulting in a reduced ability to mitigate memorization. While dense dummy sets consistently yield better performance than sparse ones, they require significantly higher computational resources during training. This presents a trade-off between reducing memorization and maintaining generalization error.

### 4.5    The label distribution of trained dummy set

We present the number of samples in the dummy set for CIFAR-10, organized by sample count. Initially, we sorted the number of samples for each label (left side of Figure 3). We also compute the average number of samples for each label across our trained dummy sets(right side of Figure 3). Since the dummy set consists of soft labels, we convert them to hard labels using the argmax operation. The results are illustrated in Figure 3. Although our initialization of the dummy set begins with randomly generated soft labels, training modifies the dummy set in a way that helps reduce the memorization of the training set. Our training procedure, as described in alg. 2, transforms the dummy set to facilitate easier memorization by the model.

## 5    Conclusion

In this work, we propose an over-memorization method that utilizes a set of redundant, meaningless instances to reduce the memorization of actual training data. By training the model with a combined set of the actual training data and a dummy set—without externally modifying the training algo­rithm—we enhance the training data privacy. Training the dummy set ensures that the dummy set is easily memorable by deep neural networks, encouraging the model to memorize these instances instead of the actual training data. We validated our approach on the CIFAR-10 image classification

task and the Wikitext-103 causal language modeling task. For both tasks, our method outperformed standard training models in terms of defense against membership inference attacks. Moreover, our method mostly maintains the utility of the model, achieving an optimal balance in the privacy-utility trade-off. We believe our approach opens a door for controlling memorization by strategically leveraging the training set.

**Reproducibility statement.** All the implementation details are described throughout 4. We will publish the code upon acceptance.

**Ethics statement.** Our work introduces the method to reduce memorization over training set by jointly training the model with pretrained dummy set. Our method supports to protect the training data privacy by decreasing the memorization within the reasonable computational cost. Our method also allows to use sparse dummy set which mainly aims to reduce computational cost.

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

## A  DERIVATION FOR EQUATION 3

$$
\begin{aligned}
\texttt{mem}(\boldsymbol{S}_t \cup \boldsymbol{S}_d, \boldsymbol{z}_t) - \texttt{mem}(\boldsymbol{S}_t, \boldsymbol{z}_t) &= \Big( \mathbb{E}_{\theta \leftarrow \mathcal{A}(\boldsymbol{S}_t \cup \boldsymbol{S}_d)} \ell_h\big(\boldsymbol{z}_t; \theta\big) - \mathbb{E}_{\theta \leftarrow \mathcal{A}(\boldsymbol{S}_t \cup \boldsymbol{S}_d \setminus \{\boldsymbol{z}_t\})} \ell_h\big(\boldsymbol{z}_t; \theta\big) \Big) \\
&\quad - \Big( \mathbb{E}_{\theta \leftarrow \mathcal{A}(\boldsymbol{S}_t)} \ell_h\big(\boldsymbol{z}_t; \theta\big) - \mathbb{E}_{\theta \leftarrow \mathcal{A}(\boldsymbol{S}_t \setminus \{\boldsymbol{z}_t\})} \ell_h\big(\boldsymbol{z}_t; \theta\big) \Big) \\
&= \Big( \mathbb{E}_{\theta \leftarrow \mathcal{A}(\boldsymbol{S}_t \cup \boldsymbol{S}_d)} \ell_h\big(\boldsymbol{z}_t; \theta\big) - \mathbb{E}_{\theta \leftarrow \mathcal{A}(\boldsymbol{S}_t)} \ell_h\big(\boldsymbol{z}_t; \theta\big) \Big) \\
&\quad - \Big( \mathbb{E}_{\theta \leftarrow \mathcal{A}(\boldsymbol{S}_t \cup \boldsymbol{S}_d \setminus \{\boldsymbol{z}_t\})} \ell_h\big(\boldsymbol{z}_t; \theta\big) - \mathbb{E}_{\theta \leftarrow \mathcal{A}(\boldsymbol{S}_t \setminus \{\boldsymbol{z}_t\})} \ell_h\big(\boldsymbol{z}_t; \theta\big) \Big) \\
&= \texttt{infl}(\boldsymbol{S}_t \cup \boldsymbol{S}_d, \boldsymbol{S}_d, \boldsymbol{z}_t) - \texttt{infl}(\boldsymbol{S}_t \cup \boldsymbol{S}_d \setminus \{\boldsymbol{z}_t\}, \boldsymbol{S}_d, \boldsymbol{z}_t).
\end{aligned}
$$

## B  DETAILED EMPIRICAL SETTINGS

**CIFAR-10,100**  For training all the models, we use SGD optimizer with learning rate of 0.1 and decaying at 30,60,90 epochs with a decaying rate of 0.5. For training dummy set, we use Adam optimizer with a learning rate of 0.01. All the models and dummy sets are trained for 100 epochs.

**Wikitext-103**  For language models, we set a learning rate of 3e-4 across all the models and use same learning rate for training dummy sets. All the models and dummy set use Adam optimizer. We concatenate the batch dummy sequence after the batch training sequences. We consistently use a batch size of 8.

