# OpenReview forum: "Preventing Unintended Memorization by Covering with Over-Memorization"
_ICLR.cc/2025/Conference — Submitted to ICLR 2025_

### Official Review · Reviewer_hz89 · 2024-10-21

**Soundness:** 2
**Presentation:** 3
**Contribution:** 1
**Rating:** 3
**Confidence:** 4

**Summary:**

The paper introduces a new approach to prevent the model from memorizing sensitive training data by introducing a dummy set—a collection of redundant, non-sensitive instances. The model is trained on both the actual training set and this dummy set. Since neural networks have limited memorization capacity, the dummy set absorbs most of the memorization, reducing the leakage of sensitive data.

**Strengths:**

- The topic of overcoming neural network overfitting and unintended memorization is a critical area.
- The paper presents multiple experiments across diverse tasks (e.g., image classification and language modeling) to evaluate the effectiveness of the proposed over-memorization method.
- The presentation is clear and well-organized, making it easy to follow.

**Weaknesses:**

1. The manuscript overlooks important related work on mitigating neural network memorization, which is the key focus of the paper.

2. While the paper presents an interesting approach using dummy data to prevent memorization, there are studies working on introducing dummy data during training to overcome overfitting by inserting (such as [2-4] and some other citations mentioned in [1]). The authors should clarify how their method advances these existing approaches.

3. While the authors evaluated training data privacy using LiRA-based MIA methods with metrics such as TPR@FPR, per-sample evaluations are missing, such as the mean success rate of the attack. It is crucial to determine whether the proposed methods can effectively defend against attacks targeting easily memorized samples.

4. The authors use randomly initialized samples as dummy data, optimized via SGD. However, this approach could raise concerns about model utility. It would be beneficial to provide further discussion and experimental results to assess the potential impact on utility.

[1] Wei, Jiaheng, et al. "Memorization in deep learning: A survey." arXiv preprint arXiv:2406.03880 (2024).

[2] Lin, Runqi, Chaojian Yu, Bo Han, and Tongliang Liu. “On the Over-Memorization During Natural, Robust and Catastrophic Overfitting.” arXiv, 13 Oct. 2023, arxiv.org/abs/2310.08847.

[3] Arpit, Devansh, et al. “A Closer Look at Memorization in Deep Networks.” arXiv, 2017, arxiv.org/abs/1706.05394.

[4] Zhang, Chiyuan, et al. “Counterfactual Memorization in Neural Language Models.” Advances in Neural Information Processing Systems, vol. 36, 2023.

**Questions:**

see above

---

> ### Author Response · Authors · 2024-11-26
> **Author response**
>
> **1. Difference from the previous works and ours**
>
> Our work presents extensive empirical results demonstrating the effectiveness of using a dummy set to decrease memorization. Additionally, we propose a novel training algorithm for optimizing the dummy set. Our method builds on existing approaches to further reduce memorization. The suggested related work will be included in the final version of this paper.
>
> **2. Per-sample evaluation of membership inference**
>
> Membership inference attacks are performed on individual samples by applying a threshold to the classification loss, where the choice of the threshold value plays a critical role. In our work, we conduct membership inference across multiple thresholds and compute the True Positive Rate (TPR) and False Positive Rate (FPR) for various threshold values, following the methodology of prior works [1]. The results for individual samples are summarized in Table 1, which reports the average memorization scores.
>
> **3. Raise concerns about model utility**
>
> The results demonstrate that our method consistently maintains performance similar to training without the dummy set. We believe this indicates that our method preserves utility while effectively reducing memorization. If by "utility" you are referring to a different concept, please let us know.
>
> [1] Carlini et al.,2021, Membership inference attacks from first principles.

---

### Official Review · Reviewer_Yqxa · 2024-10-29

**Soundness:** 2
**Presentation:** 3
**Contribution:** 2
**Rating:** 5
**Confidence:** 4

**Summary:**

This paper proposes an interesting method to decrease model memorization by introducing an additional dummy set composed of redundant and non-sensitive samples. The approach involves optimizing this dummy set on a separate model. Specifically, the dummy set is designed to fulfill two requirements: decreasing generalization error and avoiding correlation with training samples. Several experiments are conducted to demonstrate the efficacy of the proposed method on both image and text data.

**Strengths:**

1. The topic of decreasing memorization by introducing an additional dummy set is both interesting and significant in the field of machine learning.

2. The authors conduct several experiments to validate the performance of their method, including tests on various datasets encompassing image and text data.

**Weaknesses:**

1. When generating the dummy set, only the generalization error is controlled. The correlation between dummy samples and training samples is not adequately addressed, which may lead to leakage of training data through the dummy samples. A detailed analysis of these correlations would enhance the paper.

2. The paper lacks comprehensive experiments on potential side effects of the generated dummy set. For instance, investigating how the dummy set affects the model's robustness, prediction uncertainty, or fairness would provide valuable insights.

3. The performance of the model is affected by the size of the dummy set, which may limit the practicality of the proposed method in real-world applications.

4. There is no theoretical guarantee provided regarding the reduction of memorization after employing the dummy set.

5. There are some typos and inconsistencies in the manuscript, such as the notation inconsistency between $S_t \cup S_d \backslash \\{z_t\\}$ and $S_t \cup S_d \backslash z_t$.

**Questions:**

1. As claimed in the paper, the dummy set can occupy the network’s memorization capacity. Could you provide additional results to support this claim? For example, presenting memorization scores or performing Membership Inference Attacks (MIA) on both training samples and dummy samples to observe their differences would strengthen your argument.

2. Could you include some figures of the final obtained dummy samples? It would be interesting to see whether they resemble natural images or appear more like random noise.

3. In line 200, you state that “adding $S_d$ to $S_t \cup S_d$ should not cause a significant change.” Is this a typo, and should it instead be “adding $S_d$ to $S_t $” ? Also, please provide additional details on why this should hold and whether this is a strong assumption for your method.

4. During model training, did you employ data augmentation techniques? If not, what would be the effect of applying them on memorization?

5. Please provide a clear definition of unintended memorization and clarify the differences between normal memorization.

6. What is the number of reference models used in the Membership Inference Attacks, and how do you choose the data used to construct these reference models?

---

> ### Author Response · Authors · 2024-11-26
> **Author response**
>
> **1. Correlation with the dummy set and the training set**
>
> We compute the influence score of the dummy set on each training sample, and the results show that the score is near 0. This suggests that extracting knowledge from the dummy set does not compromise the privacy of the original training samples.
>
> **2. The model's performance is affected by the size of the dummy set.**
>
> Using the dummy set results in a slight degradation in performance while safeguarding the privacy of the training set. This behavior reflects the utility-privacy trade-off, where our method successfully achieves an optimal balance between the two.
>
> **3. Theoretical guarantee of our method**
>
> In Table 1, we present the memorization scores calculated using the memorization estimator proposed in [1]. This estimator provides low-variance estimates, making it suitable for precise evaluation, and is directly employed in our analysis.
>
> [1] Feldman et al. 2020, What neural networks memorize and why: discovering the long tail via influence estimation.

---

### Official Review · Reviewer_zVD7 · 2024-11-01

**Soundness:** 1
**Presentation:** 3
**Contribution:** 2
**Rating:** 3
**Confidence:** 4

**Summary:**

This paper proposes the use of a dummy training set to reduce privacy leakage from model memorization. The dummy set is initialized with random noise, and beyond training the model to memorize this set, the authors introduce an additional step: optimizing the dummy set itself to enhance memorization. Experimental results indicate that training with this dummy set can mitigate privacy leakage, as measured by membership inference attacks.

**Strengths:**

1. The idea of using a dummy set to counteract the memorization of private data is novel and intriguing.

2. The paper is well-organized and presented clearly.

**Weaknesses:**

Weaknesses:

1. Optimizing the dummy set itself to facilitate memorization may counteract its ability to consume model capacity effectively. The authors claim that this optimization can "improve generalization performance" (line 248). However, this assertion is neither supported by experimental evidence nor intuitively clear, as it is not evident why optimizing the loss of the dummy set would lead to better generalization.

Follow-up questions:

(1) Can the authors provide a comparison of membership inference attack accuracy with and without dummy set optimization?

(2) Can the authors report the test accuracy results with and without this optimization?

2. The reductions in memorization scores shown in Table 1 are not substantial. While it is commendable that the authors report standard deviations, the variability appears larger than the observed decrease in scores.

3. The improvement in resistance to membership inference attacks is not pronounced. For CIFAR-10, the AUROC only decreases from 0.6373 to 0.5995, while for Wikitext-103, it drops from 0.9688 to 0.7972.

Follow-up question: What would the attack AUROC be when employing conventional regularization techniques such as dropout or label smoothing?

4. How would the optimal size of the dummy set scale with model size? For instance, would larger models necessitate proportionally larger dummy sets?

**Questions:**

Please see weaknesses.

---

> ### Author Response · Authors · 2024-11-26
> **Author response**
>
> **1. Evidence for generalization while using the optimized dummy set - test accuracies, membership inference results with and without dummy set optimization.**
>
> We briefly compare the performance with and without dummy set optimization. When the dummy set optimization is not applied, the model achieves test accuracies similar to those shown in Table 2. The memorization scores for both cases are presented in Table 1.
>
> **2. The variance of the memorization score in Table 1 is big.**
>
> Since each sample in the training set shows high variance respect to the memorization score, the variance in Table 1 is big. For instance, the biggest value of memorization score with respect to the loss measure (mem-loss) shows 15.544 where the smallest one shows near 0. For the memorization score based on predictions (mem-pred), the biggest memorization score was 1 where the smallest one shows near 0. Thus, it is natural to observe high variance in memorization score.
>
> **3. The optimal size of the dummy set and its corresponding hyperparameters.**
>
> We believe the optimal configuration depends on the specific task being addressed. Based on the results in Tables 2 and 4, a dummy set with a size equal to one-fifth of the training set performs best in terms of both memorization reduction and overall performance. However, for causal language modeling, the dummy set with the smallest sparsity value yields the best results for both memorization and performance.

---

### Official Review · Reviewer_LobW · 2024-11-06

**Soundness:** 2
**Presentation:** 3
**Contribution:** 2
**Rating:** 3
**Confidence:** 3

**Summary:**

This paper designs a new approach to mitigate the privacy-utility trade-off of ML models. Specifically, the authors propose to first synthesize a dummy dataset that can simulate the properties of the original training dataset, and then train ML models on this synthesized dummy dataset. The authors claim that since the dummy dataset does not contain original training samples, thus it can protect the privacy of original training data. Besides, since the dummy dataset can simulate the original training dataset well, models trained on it can also gain strong enough generalizability. Experiments are conducted on the image classification task and the text generation tasks and evaluated with a memorization metric and a membership inference attack (i.e., LiRA).

**Strengths:**

1. This paper is well-written and easy to follow.

2. Experiments are conducted not only on simple image classification tasks but also on language generation scenarios, which I appreciate.

**Weaknesses:**

1. I have two main concerns related to "dataset condensation" [r1]:
    - The idea of using synthesized dummy datasets to simulate real-world datasets is very similar to the concept of "dataset condensation" (or say "dataset distillation") which first appeared in 2021 [r1]. See also a survey [r2] for more details. So I think the authors should discuss what is the difference between their dummy dataset method and dataset condensation and also add several recent dataset condensation methods as baselines for comparison [r1, r3, r4].

    - Another main concern is using a synthesized training dataset (or say using "dataset condensation") to preserve privacy has already been studied in 2022 by [r4]. Worse still, [r5] further shows that [r4] might not be able to protect data privacy as the method by [r4] does not have correct DP-based privacy accounting and simple privacy attacks can easily defeat it. Given that the proposed dummy dataset method does not even have any privacy accounting but only empirical analysis with privacy attacks, I don't have enough confidence that it can indeed enjoy a strong privacy-preserving ability (since it may be defeated by another privacy attack as it does not have any theoretical privacy-preserving guarantee). Please comment.

2. In the experiments, since the proposed dummy dataset method is a kind of privacy-preserving method, I think in Table 2 and Table 3 the authors should compare their proposed method with DP-SGD baselines.

3. I think the authors should add experiments similar to Table 2 and Table 3 for the CIFAR-100 dataset.

4. In Line#4 of Algorithm 2, why do you need to update the model parameter $\tilde \theta$ with gradients calculated on the dummy set $S_d$? I think it is necessary to include an intuitive explanation and an ablation study on this gradient term.

5. In Table 2, why the test accuracy of clean-trained ResNet-18 on the CIFAR-10 dataset is as low as 87.58%? To the best of my knowledge, one can easily train a ResNet-18 on CIFAR-10 that achieves a test accuracy of over 90%. Please comment.

6. Again, although the authors claim that the dummy trainset can protect privacy, I think such a claim is incorrect and will only provide a false sense of security to the ML S&P community. The main flaw is that the proposed method is just an empirical method and does not have any theoretical privacy-preserving analysis/guarantee. [r5] has already shown how easy it is to break a privacy protection method that does not enjoy strong theoretical privacy-preserving accounting. Maybe this paper can focus more on explaining ML memorization rather than protecting ML privacy.

**References:**

[r1] Zhao B. et al. Dataset Condensation with Differentiable Siamese Augmentation. ICML 2021.

[r2] Lei S. et al. A comprehensive survey of dataset distillation. TPAMI 2024.

[r3] Zhao B. et al. Dataset Condensation with Gradient Matching. ICLR 2021.

[r4] Dong T. et al. Privacy for Free: How does Dataset Condensation Help Privacy? ICML 2022.

[r5] Carlini N. et al. No Free Lunch in "Privacy for Free: How does Dataset Condensation Help Privacy". arXiv:2209.14987 (Technical Report).

**Questions:**

See **Weaknesses** for details.

---

> ### Author Response · Authors · 2024-11-26
> **Author response**
>
> **1. Difference between the dummy set and the dataset from condensation.**
>
> Unlike dataset condensation, the dummy set is not constrained in its visibility (for example, its pixel values are not limited to the range 0 to 1; most of our dummy images have pixel values ranging from -40 to +40). Additionally, we compute the influence score of the dummy set on each training sample, which is found to be near 0. This indicates that extracting knowledge from the dummy set does not compromise the privacy of the original training samples.
>
> **2. Updating model parameter $\tilde{\theta}$ in Algorithm 2**
>
> In Algorithm 2, the model parameter $\tilde{\theta}$ is exclusively utilized for training the dummy set. Consequently, without updating $\tilde{\theta}$, the model $\tilde{h}_{\tilde{\theta}}$​ cannot function as a valid model for the given task. The primary objective is to optimize the dummy set to reduce memorization of the training set in deep neural networks. Therefore, it is crucial to update both $\tilde{\theta}$ and the dummy set during this process. After training the dummy set using Algorithm 2, the parameter $\tilde{\theta}$ is no longer employed in the subsequent steps. Instead, a new model is trained, which aims to minimize memorization of the training set.
>
> **3. Lower accuracy of CIFAR-10 in Table 2.**
>
> The objective of Table 2 is to demonstrate that training with the dummy set effectively defends against membership inference attacks. Each model is trained using a dataset of size 25,000 (as mentioned in line 301) to validate the defense method against membership inference. Both the reference models and the models trained with the dummy set achieve comparable test set accuracy, as presented in Table 2.

---

### Meta-Review · Area_Chair_ibfY · 2024-12-21

**Metareview:**

The paper introduces an approach to mitigate privacy leakage caused by model memorization by utilizing a dummy training set. Initially, the dummy set is generated with random noise. Beyond training the model to memorize this set, the authors incorporate an additional optimization step to improve the dummy set's memorization efficiency. Experimental results demonstrate that using this optimized dummy set reduces privacy leakage, as evaluated through membership inference attacks.

The paper is easy to follow, but has several weak points. For example, as one of the reviewers has pointed out, the privacy preservation effort by using a a synthesized training dataset has been considered in the past, and also has been shown that it is not very effective. The paper fails to provide a comprehensive comparison with those previous work. Overall, the main claim of the paper -- using the dummy set can preserve privacy -- seems to not have strong evidence in the paper. Therefore, the decision of the paper is Reject.

**Additional Comments On Reviewer Discussion:**

Most of the reviewers had negative opinion about the paper.

Especially, Reviewer LobW pointed out that the usage of dummy set is shown to be not very effective in previous work, and the authors could not appropriately rebut to the comment.

Reviewer zVD7, hz89, and Yqxa also pointed out the paper misses comparing with important baselines on mitigating memorization, etc. and the authors only provided short rebuttal that did not fully address the concerns of the reviewers.

---

### Decision · Program_Chairs · 2025-01-22

Reject